# Overexpression of the *Caragana korshinskii com58276* Gene Enhances Tolerance to Drought in Cotton (*Gossypium hirsutum* L.)

**DOI:** 10.3390/plants12051069

**Published:** 2023-02-27

**Authors:** Yuanchun Pu, Peilin Wang, Jiangling Xu, Yejun Yang, Ting Zhou, Kai Zheng, Xinwu Pei, Quanjia Chen, Guoqing Sun

**Affiliations:** 1College of Agronomy, Xinjiang Agricultural University, Urumqi 830052, China; 2Biotechnology Research Institute, Chinese Academy of Agricultural Sciences, Beijing 100081, China; 3College of Agronomy, Shanxi Agricultural University, Jinzhong 030801, China

**Keywords:** overexpression, *Caragana korshinskii*, *com58276*, drought tolerance, cotton

## Abstract

The increasing water scarcity associated with environmental change brings significant negative impacts to the growth of cotton plants, whereby it is urgent to enhance plant tolerance to drought. Here, we overexpressed the *com58276* gene isolated from the desert plant *Caragana korshinskii* in cotton plants. We obtained three OE plants and demonstrated that *com58276* confers drought tolerance in cotton after subjecting transgenic seeds and plants to drought. RNA-seq revealed the mechanisms of the possible anti-stress response, and that the overexpression of *com58276* does not affect growth and fiber content in OE cotton plants. The function of *com58276* is conserved across species, improving the tolerance of cotton to salt and low temperature, and demonstrating its applicability to improve plant resistance to environmental change.

## 1. Introduction

Water is an important environmental factor that affects crop growth and limits the geographical distribution of plants. Global climate warming is accompanied by water scarcity, which constitutes a serious threat to agricultural production [1,2]. Cotton (*Gossypium hirsutum*) is an important and global crop that is valued for its renewable fiber resources. In China, cotton is also an important oil crop and the main source of natural fiber, representing an important resource for national economy and security [3]. Since the 1980s, an aggravating drought trend in China has reduced the total annual cotton yield by more than 30% [4]. Since more than half of the production of cotton worldwide is conducted in severely water-scarce areas, drought has become a major and increasing concern to the sustainable development of cotton crops [5], and an important limiting factor affecting yield and fiber quality. Hence, the cultivation and development of new varieties of drought-resistant cotton constitutes an important strategy to enhance the harvest of grain and cotton [6].

Drought affects cotton growth by decreasing the plant height, total biomass, and leaf area [7]. In addition, during the flower boll stage, drought can significantly impact the number of branches, fruit nodes, split bolls, and total bolls present. The cotton varieties with poor drought tolerance also have higher pollen sterilization rates. Drought mostly affects the plant at the surface, since the underground root system can extend deeper into the soil to reach water sources [8]. This manifests through the inhibition of growth of the main stems and fruit branches, and partially or completely closed leaf stomata that lead to a decrease in the photosynthetic rate and the inhibition of mineral and nutrient transport. Moreover, the young part of the plant wilts and the bud shedding increases, resulting in a decrease in lint yield and quality that ultimately leads to plant death. The period between the budding and the flowering stages is when cotton is more sensitive to drought stress, which inhibits the differentiation of male and female organs and pollen development and subsequently causes poor pollination rates, high seed sterilization rates, reduced effective boll numbers, small buds, and a reduced single boll weight [9]. Drought can also cause premature aging of cotton plants, resulting in a decrease in the photosynthetic rate of cotton leaves, a shortening of their functional period, a reduction in the expansion of cotton bolls, and a decrease in sub-fingers and fiber quality. Other results have further shown that the specific strength of cotton fiber after short-term drought treatment is significantly reduced compared with plants grown under control conditions [10,11].

*Caragana korshinskii* (*C. korshinskii*), commonly known as peashrub, is a small shrub that belongs to the botanical family Fabaceae and is widely distributed across the sandy grasslands of Northwestern China and Mongolia [12]. Due to the long-term arid and saline environment it inhabits, the peashrub has evolved under selective constraints over long periods of time, which allowed it to adapt to extreme drought and salt conditions [13,14]. In addition, it blooms and flourishes, making it an excellent nectar source. It is an important tree for windproof and sand-fixing forest and soil and water conservation forest. *C. korshinskii* also has a strong environment adaptability, with both cold and high temperature resistance. In the average annual temperature of 1.5 °C, the lowest temperature of −42 °C, and the maximum depth of the frozen soil of 290 cm, peashrub can remain safely intact over the winter. Previous studies have identified several functionally important genes in *C. korshinskii*, including CkLEA1, a gene involved in multiple stress responses that is highly expressed under drought, ABA, cold, heat, and salt stress conditions [15]. In addition, real-time-quantitative PCR analysis showed that drought stress enhances CkWRKY1 expression, indicating that this gene might be involved in drought resistance [16] and play an important role in the biosynthesis and regulation of ABA [17]. Long et al. (2020) [18] performed transcriptome sequencing of *C. korshinskii* under drought treatment and demonstrated that the *com58276* transcript plays a role in drought resistance.

While increasing research efforts were employed to identify drought-resistant genes in cotton, these results were not translated into functional traits for the development of new drought-resistant varieties. Here, we evaluate the role played by *com58276* in promoting resistance to drought stress in cotton plants. Specifically, we transferred the exogenous gene *com58276* from *A. tumefaciens* into the cotton genome, screened drought resistance in the transgenic strains, and analyzed the molecular mechanisms of drought resistance promoted by *com58276* using RNA-seq. Our results provide insightful ideas and technical support for the development of new drought-resistant cotton varieties with importance for production and cultivation of this economically important crop in dry environments.

## 2. Results

### 2.1. Gene Cloning and Subcellular Localization

The root system of the drought-tolerant desert crop is well developed (Figure 1A), which help it can grow well in an extreme water shortage environment. In a previous study, the *com58276* gene was identified by DEG analysis of the transcriptome after drought treatment, which encodes for 84 amino acids and includes 46 bp in 5*′*UTR and 11 bp in 3*′*UTR (Figure 1B) regions. After extracting RNA and reversing transcription into cDNA, and using cDNA as template for amplification, we cloned the complete *C. korshinskii* transcript by RT-PCR (Figure 1C). Interestingly, a Basic Local Alignment Search Tool (BLAST) analysis on cottonFGD (https://cottonfgd.net/ (accessed on 1 September 2022)) and the National Center for Biotechnology Information (NCBI) found no homologous genes in cotton and other plants, nor similar DNA or protein sequences, demonstrating that this is a *C. korshinskii*-specific gene. We further constructed a genetic transformation vector (Figure 1D) and a subcellular localization vector (Figure 1E), in which *com58276* was fused with the enhanced green fluorescent protein eGFP. The subcellular localization results showed that the COM58276 protein was localized to the cell membrane and the cytoplasm (Figure 1F).

### 2.2. Genetic Transformation and Positive Identification

We obtained four positive cotton plants by DsRed2 fluorescence and kanamycin resistance screening at the calluses stage through *Agrobacterium* infection. There were no significant differences in seedling phenotype compared with controls (TM-1, a conventionally cultivated cotton material) (Figure 2A). To avoid confounding false positives, we further analyzed four overexpression *com58276* cotton plants by RT-PCR, of which only three were positive (OE2 was negative for the *com58276* gene) (Figure 2B). The amplification product was sequenced and aligned to the *com58276* reference sequence. qRT-PCR showed OE4 had the highest RNA expression levels (Figure 2C). A northern blot showed that *com58276* was properly expressed in the three OE plants’ leaves (Figure 2D).

### 2.3. Overexpression of the com58276 Gene Enhances Drought Tolerance in Cotton

In order to further determine whether *com58276* enhances drought tolerance in cotton, we used 5% PEG6000 solution to soak WT and OE seeds. We found that the number of seeds germinated by OE1, OE3, and OE4 cotton were significantly higher than WT (Figure 3A). We divided WT and OE1, OE3, and OE4 into 10 groups, each containing 20 seeds, and further evaluated germination rates before and after treatment. We found that the germination rate of WT, OE1, OE3, and OE4 were identical at 18–20 before treatment (Figure 3B). In contrast, we found that only 6–9 WT seeds germinated after treatment, while OE plants showed 11–16 germinations, illustrating a significant difference between the groups (Figure 3C). The same results were obtained for natural drought treatment at the seedling stage, in which we stopped watering the seedlings to make them naturally dry. Moreover, WT plants showed wilting symptoms after one week of water shortage, but no similar phenotypes were observed in OE plants (Figure 4A). We measured plant height and relative water content (RWC) separately, and found the former was significantly lower in WT compared with OE following drought stress (Figure 4B), while RWC was 50–55% significantly lower than OE (Figure 4C). We further scanned the root systems and found fewer lateral roots in WT than OE (Figure 4D), as well as a significantly lower weight aboveground (Figure 4E) and underground (Figure 4F) in WT. Interestingly, the highest expressed OE4 showed a better drought tolerance.

When plants are stressed, the physiological and biochemical indexes in the body will change, and the amount of index change can be calculated to measure the degree of plant resistance. SOD (superoxide dismutase) and CAT (catalase) activity, and Pro (proline) and MDA (micro malondialdehyde) content are important physiological indicators of plant stress tolerance. We found no differences in these indicators under normal conditions between WT and OE plants. In contrast, we found significant differences in Pro content between WT (38.07 μg/g·FW) and transgenic plants (OE1, OE3, and OE4 with 56.85, 52.52, and 66.41 μg/g·FW, respectively) (Figure 5A). Similarly, CAT enzyme activity was significantly higher in OE plants (OE1, OE3, and OE4 with 42.96, 47.89, and 56.01 U·g^−^^1^, respectively) compared with WT (26.13U·g^−^^1^) (Figure 5B). We observed similar patterns in SOD enzyme activity, with WT (615.04 U·g^−^^1^) showing significantly lower values than OE plants (OE1, OE3, and OE4 with SOD values of 824.48, 790.67, and 806.06 U·g^−^^1^, respectively; Figure 5C). Finally, the MDA content of WT plants (38.08 nmol·g^−^^1^) was significantly higher than OE plants (OE1, OE3, and OE4 with 20.86, 22.52, and 19.41 nmol·g^−^^1^, respectively; Figure 5D). Similarly, the highest expressed OE4 showed a better drought tolerance.

### 2.4. Transcriptome Analysis Reveals the Mechanisms of Drought Tolerance Induced by com58276

Next, we employed transcriptome sequencing of WT and OE cotton plants after drought stress. The data after the completion of database sequencing were analyzed by Principal Component Analysis (PCA). PCA showed that the data of WT and OE were well clustered. There are 32,763 genes up-expressed and 31,952 down-expressed in OE1, 33,663 genes up-expressed and 31,073 down-expressed in OE3, and 34,256 genes up-expressed and 31,180 down-expressed in OE4. To further determine the accuracy of transcriptome data, we randomly selected four up-regulated (GH_A11G0086, GH_D07G1191, GH_A11G1547, and GH_D05G2806) and four down-regulated (GH_D12G2894, GH_A13G2130, GH_A13G1943, and GH_D10G1844) genes in DEGs of the transcriptome for qRT-PCR analysis. The expression trends of the two sets were consistent, indicating the accuracy of the transcriptome results (Figure 6A).

In order to ensure the accuracy of analysis, we screened DEGs by fold-change and *p*-value and selected common genes from the genes screened out WT vs. OE1, WT vs. OE3, and WT vs. OE4 for further GO and KEGG analysis.We performed GO and KEGG analyses on DEGs. GO enrichment analysis showed that microtubule-related terms were involved, such as tubulin binding, microtubule binding, microtubule motor activity, microtubule-based process, and microtubule-based movement. These are highly correlated with the maintenance of cell morphology under drought stress and water transport within helper cells, under which different plants show different response strategies (e.g., sea buckthorn increases drought resistance by reducing the diameter and increasing the density of the duct). In addition, some carbohydrates were enriched for anabolic processes, such as cellular glucan metabolic process, cellular carbohydrate biosynthetic process, carbohydrate biosynthetic process, and cellular carbohydrate metabolic process. Interestingly, many photosynthesis-related terms were also enriched, including photosystem, photosystem Ⅱ, oxidoreductase complex, photosystem Ⅱ oxygen evolving complex, and photosynthetic membrane (Figure 6B). Photosynthesis is an important metabolic process in plants that is very sensitive to water loss. Under drought conditions, the strength of photosynthesis has a strong impact on external plant growth, yield, and quality and decreases the photosynthetic rate.

KEGG enrichment analysis showed enrichment for ABC transporters, a family of proteins widely present in plants which are highly associated with drought and that play an important role in physiological plant processes, including excretion of heavy metal ions and resistance to drought stress. In addition, a large number of amino acid metabolic pathways were enriched, such as tyrosine metabolism, phenylalanine metabolism, and glycine, serine, and threonine metabolism. Among these, phenylalanine promotes the synthesis of lignin; glycine has a unique effect on the photosynthesis of crops, which is conducive to crop growth and increases the amount of sugar; threonine improves tolerance and insect pest damage; and tyrosine increases plant drought tolerance. In addition, some metabolic sugar pathways, such as fructose and mannose metabolism and starch and sucrose metabolism, were also enriched (Figure 6C), suggesting different drought response mechanisms mediated by *com58276*. Overall, the transcriptome results under drought stress showed that OE plants exhibit greater growth and physiological changes, and faster and more intense responses. These results provide a reference for further study of plant adaptation mechanisms under drought stress.

### 2.5. Overexpression of com58276 Has No Negative Effects on Cotton

In order to further determine whether transgenic cotton could have negative effects on production, we measured and analyzed statistics on cotton plant type structure in the middle growth stage and fiber yield and quality in the later receiving stage and compared these with WT to determine whether there was a statistical difference. We measured the number of sections, the number of fruits, and the plant height between OE and WT TM-1 cultivars at the flowering stage but found no significant differences between them (Figure 7A–C). These data demonstrate that the *com58276* gene does not affect cotton plant growth. We also measured the yield and fiber quality; the weight of lint and unginned cotton was used as the yield evaluation standard (Figure 7D,E), and the length of fiber, specific strength of fiber, and micronaire of fiber were used as the fiber evaluation standard. Once again, no significant differences were found between OE and control TM-1 plants (Figure 7F–H), suggesting that transgenic *com58276* does not affect yield and fiber quality.

## 3. Discussion

### 3.1. The Function of com58276 Is Conserved in Plants

To determine whether *com58276* is functionally conserved in other species, we transfected the gene into tobacco. We found that overexpressed *com58276* tobacco displayed enhanced drought tolerance compared with WT tobacco, consistent with observations in *Arabidopsis thaliana*—the overexpression of *com58276* in *Arabidopsis thaliana* improved drought tolerance in a study by Long et al., 2020—and in cotton. We set the concentration gradients to 1%, 3%, and 5% PEG6000 to treat tobacco plants, and found the following: no phenotypic differences at 1% concentration (Figure 8A); WT tobacco began to wilt, but OE did not, at 3% concentration (Figure 8B); WT tobacco completely withered, while OE tobacco showed a slight wilting phenotype, at 5% concentration (Figure 8C). Pro content and CAT activity were significantly increased in OE tobacco after drought treatment, while MDA activity was significantly decreased. Pro content and CAT and MDA activity also showed that OE tobacco had a better tolerance to drought than WT plants (Figure 8D,E). In summary, we believe *com58276* is a good candidate gene for enhancing drought tolerance in plants.

### 3.2. com58276 Is Involved in a Variety of Adversity Stress Responses

From the transcriptome results, we found that most of the pathways enriched by GO and KEGG analyses are involved in adversity defense responses, not just to drought tolerance. In addition, *C. korshinskii* itself has a good adaptability, can withstand extremely low-temperature environments, and has a well-developed root system. To further explore whether *com58276* has other functions, we treated TM-1 and OE plants with salt and low-temperature stresses. We found that, after three days of treatment with 300 mM NaCl, the cotyledons of TM-1 began to show significant wilting, while OE wilted only weakly (Figure 9A). The results from qRT-PCR showed that the expression of *com58276* increased by 3.56, 4.93, and 3.98-fold in OE1, OE3, and OE4 after NaCl treatment (Figure 9B), indicating that this gene responds to salt stress and triggers a pathway resistance response. For the low-temperature treatment, the OE phenotype was also significantly improved compared with TM-1 plants, which were completely withered, while OE plants showed obvious stress tolerance (Figure 9C). The expression of *com58276* also increased by 5.56, 5.74, and 4.16-fold in OE1, OE3, and OE4, respectively (Figure 9D), indicating that this gene is also involved in low-temperature stress responses.

## 4. Materials and Methods

### 4.1. Plant Materials, Growth Conditions, and Stress Treatment

Conventional upland cotton (*Gossypium hirsutum* L.) varieties TM-1 and OE plants were grown in a mixture of vermiculite and nutrient soil (1:2, *w*/*w*) at 28 °C/25 °C (light/dark) under a 16 h photoperiod. For the drought treatment, seeds were treated with 5% PEG6000 for one day, and each treatment consisted of 20 seeds with ten replications. For the plants treatment, we ceased watering after 3 weeks of normal growth of the cotton seedlings. For the salt treatment, seedling plants were treated with 300 mM NaCl for three days. For the cold treatment, seedling plants were treated with 8 °C for three days. Leaves were then excised, immediately frozen with liquid nitrogen, and stored at −80 °C until analysis.

NC89 was used for genetically modified tobacco and controlled by wild NC89. WT and OE tobacco were grown in a mixture of vermiculite and nutrient soil (1:2, *w*/*w*) at 28 °C/25 °C (light/dark) under a 16 h photoperiod. For the drought treatment, seedling plants were treated with 1%, 3%, and 5% PEG6000 for three days. Leaves were then excised, immediately frozen with liquid nitrogen, and stored at −80 °C until analysis. 

### 4.2. Determination of Proline, SOD, and CAT Activity

Cotton leaf tissues were sampled, and the activities of three enzymes were measured after drought treatment. Enzyme activities of the control plants were also measured from seedlings grown in normal conditions (CK). Proline, superoxide dismutase (SOD), micro malondialdehyde (MDA), and catalase (CAT) activities were measured using the Solarbio Proline Assay Kit BC0290; Superoxide Dismutase Assay Kit: BC0170; Micro Catalase Assay Kit: BC0205; and Micro Malondialdehyde (MDA) Assay Kit: BC0025, Beijing, China), respectively. 

### 4.3. RNA Sequencing and Data Analysis

The treated cotton leaves were sampled and RNA extracted. Library construction and sequencing were performed in accordance with the standard experimental procedures provided by the Illumina 2000 system. The differentially expressed genes (DEGs) were identified using DESeq2 [19] by a |Fold Change| ≥ 1. The resulting *p*-value was adjusted using Benjamini and Hochberg’s approach. Genes with an adjusted *p*-value < 0.05 as determined by DESeq2 were assigned as differentially expressed. EggNOG-mapper v2 [20] was used for gene annotation. Gene Ontology (GO) and Kyoto Encyclopedia of Genes and Genomes (KEGG) pathway enrichment analysis of the DEGs was performed using the cluster profiler R package [21].

### 4.4. RNA Extraction, cDNA Preparation, and qRT-PCR Analyses

Total RNA was extracted using an RNA extraction kit (FastPure Plant Total RNA Isolation Kit) and digested with DNaseI to eliminate genomic DNA (Polysaccharides and Polyphenolics-rich, RC401-01, Vazyme). Approximately 2 μg of total RNA was reverse-transcribed (HiScript III All-in-one RT SuperMix Perfect for qPCR, R333-01, Vazyme). Quantitative real-time PCR was performed with an ABI 7500 Real-Time PCR system using ChamQ Universal SYBR qPCR Master Mix (Q711-02, Vazyme). Three plants were selected each time for mixing, and this was repeated three times. The cotton *Histone3* and *UBQ7*, with stable expression in different tissues, developmental stages, and environmental conditions [22], were used as internal controls, and the relative expression level was measured using the 2^−ΔΔCT^ method [23]. 

### 4.5. Subcellular Localization of COM58276 Proteins

The open reading frames of *com58276* were inserted into the pCAMBIA1302 vector. The stop codon of the gene was removed and expressed in tandem with the enhanced green fluorescent protein eGFP. This construct was introduced into *Agrobacterium tumefaciens* EHA105 (pSoup), which was subsequently transformed into cotton protoplasts; transformation was carried out according to the optimal concentration and time [24]. The protoplast was analyzed by confocal microscopy (laser confocal super-resolution microscope LSM980) with bright field and fluorescence imaging. 

### 4.6. Data Recording for Morphological Traits

The cotton plants were grown in a standard cotton field at the Experimental Station of the Biotechnology Research Institute, Chinese Academy of Agricultural Sciences, Pinggu, Beijing Province. Field planting followed a randomized complete-block design with three replicates. Each tested transgenic cotton line was planted in an experimental plot. The plant-to-plant distance was 12.5 cm and the row-to-row spacing was 30 cm. All recommended plant protection measures detailed by Gwathmey et al. [25] were adopted from sowing to harvesting. One hundred plants from each repeat were randomly selected for important agronomic traits including plant height, number of fruit branches per plant, and number of bolls per plant, which were measured on a single-plant basis. Plant height was determined as the height of the main stem at the boll opening stage. Vegetative shoots and the fruiting branch of the main stem were separated manually for measurements of the number of fruit branches per plant. 

## 5. Conclusions

We showed that the overexpression of *com58276* in cotton improves plant stress tolerance to drought, salt, and low temperatures. Although the mechanisms of *com58276*-mediated stress resistance remain unclear, they are worth studying. Plant phenotypes and the determination of multiple physiological indicators throughout the growth cycle indicate that the overexpression of *com58276* has no negative effects on cotton. Therefore, we next plan to study stress tolerance mechanisms and plant resistance in complex field experiments.

## Figures and Tables

**Figure 1 plants-12-01069-f001:**
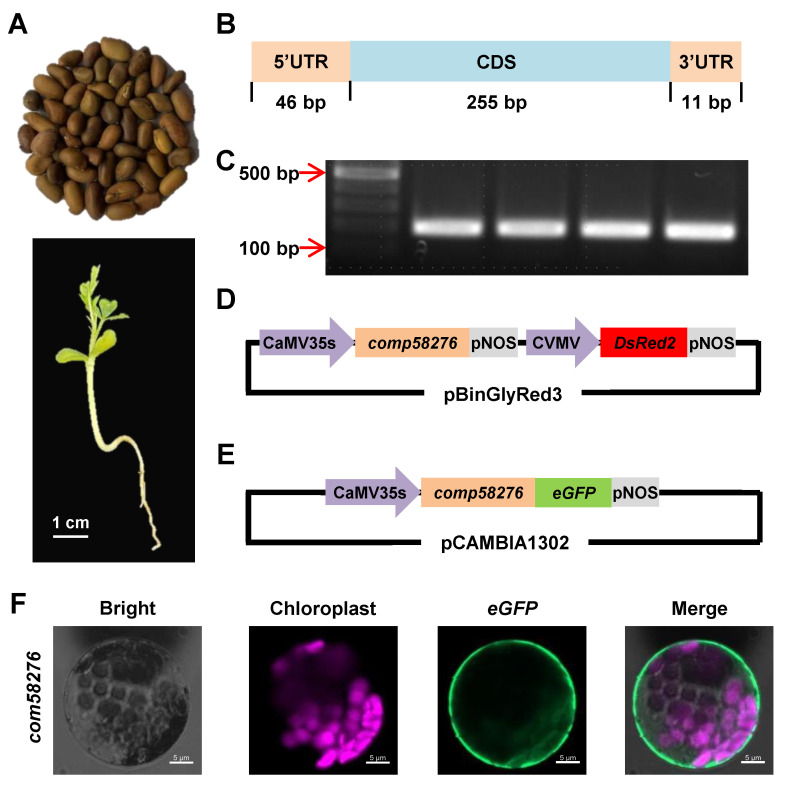
Cloning of *C. korshinskii com58276*, vector information, and subcellular localization. (**A**) *C. korshinskii* seeds and seedlings. Scale bars = 1 cm. (**B**) Gene structure. (**C**) Clone *com58276* by RT-PCR. (**D**) The *com58276* overexpression vector. (**E**) The *com58276* subcellular localization vector. (**F**) Subcellular localization assay of pCaMV35S::COM58276-eGFP in cotton protoplast. Pink fluorescence is chloroplast auto-fluorescence. Scale bars = 5 μm.

**Figure 2 plants-12-01069-f002:**
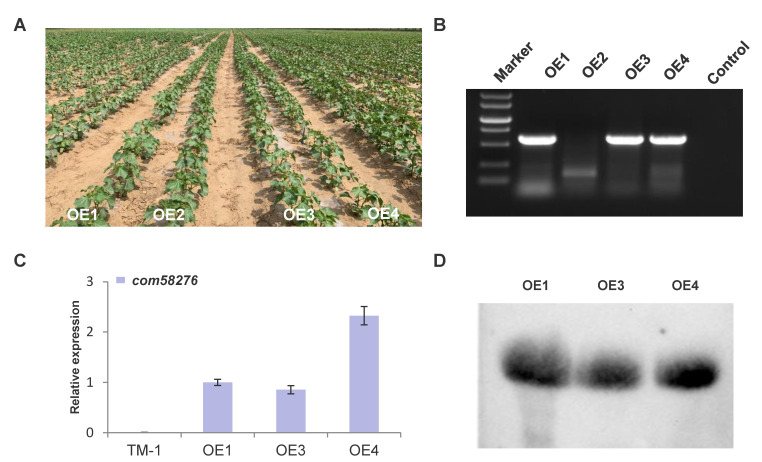
Overexpression *com58276* cotton field phenotype and identification. (**A**) The phenotype of overexpression *com58276* cotton in the field. (**B**) RT-PCR amplification results of transgenic cotton. (**C**) The qRT-PCR result of OE cotton and TM-1. (**D**) The northern blot result of OE cotton.

**Figure 3 plants-12-01069-f003:**
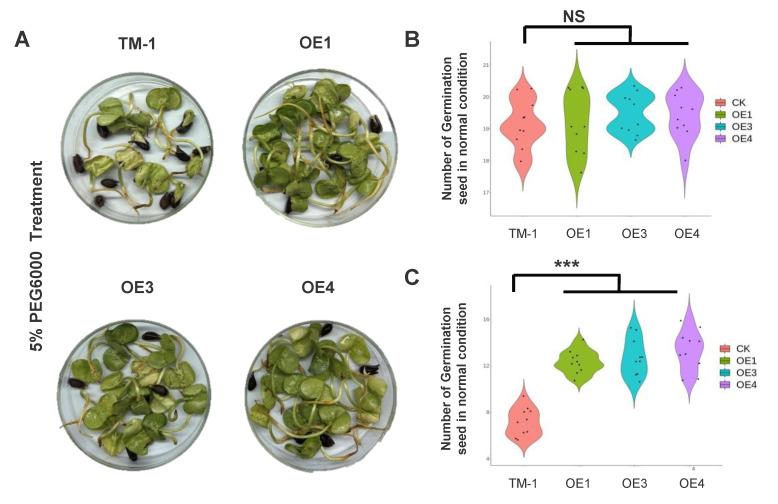
Drought treatment and germination rate of overexpression *com58276* cotton seeds. (**A**) Germination of TM-1 with OE1, OE3, and OE4 cotton seeds under 5% PEG6000 treatment. (**B**) The number of germinating seeds under normal conditions. (**C**) The number of germinating seeds after PEG6000 treatment. (NS (no significance), *p* > 0.05, *** *p* < 0.001, Dunn’s test).

**Figure 4 plants-12-01069-f004:**
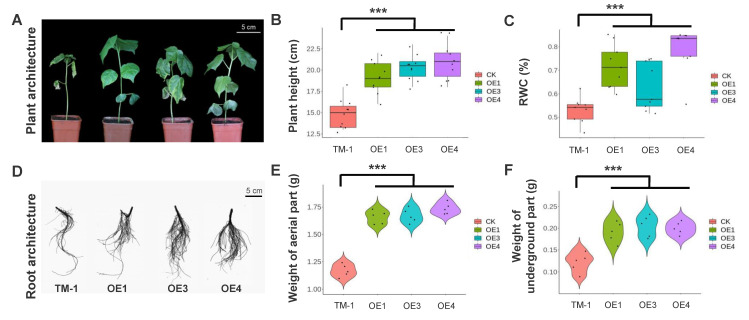
Drought treatment and physiological indicators of overexpression *com58276* cotton. (**A**) Plant architecture of TM-1 and OE1, OE3, and OE4 after drought treatment. (**B**) The plant height of TM-1 and OE1, OE3, and OE4. (**C**) The relative water content (RWC) of TM-1 and OE1, OE3, and OE4. (**D**) Root architecture of TM-1 and OE1, OE3, and OE4. (**E,F**) The weight of superficial and underground structures of TM-1 and OE1, OE3, and OE4 plants. (*** *p* < 0.001, Dunn’s test).

**Figure 5 plants-12-01069-f005:**
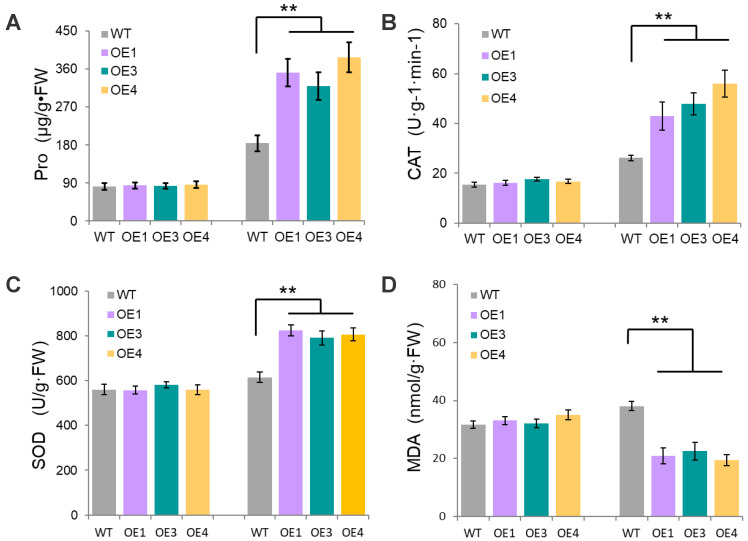
Proline, CAT, SOD, and MDA activities under treatment. (**A**) Proline. (**B**) CAT. (**C**) SOD. (**D**) MDA. Values are represented as the mean ± S.D. of three biological replicates. ** *p* < 0.01 Student’s *t* test.

**Figure 6 plants-12-01069-f006:**
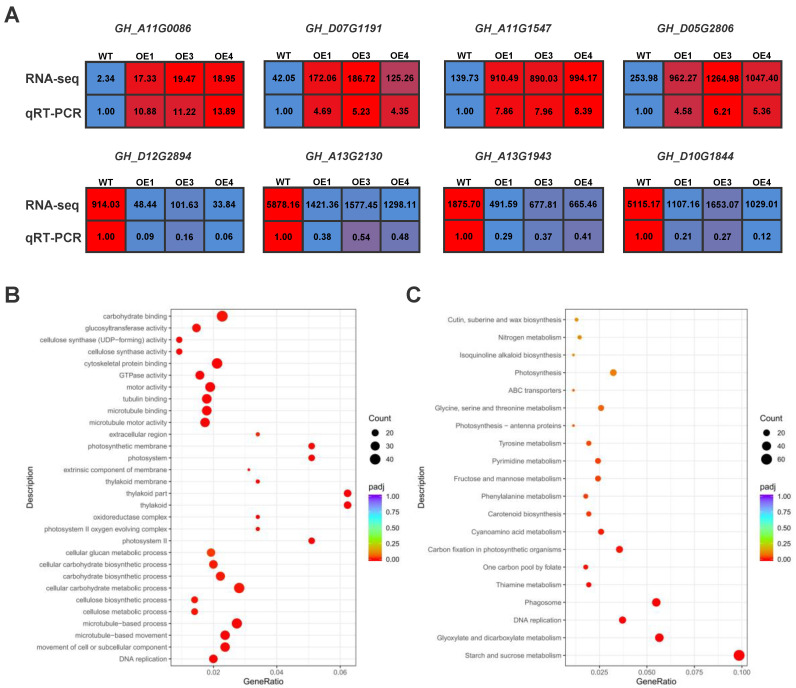
RNA-Seq analysis of OE1, OE3, and OE4 after drought treatment. (**A**) qRT-PCR validation of four up- and four down-regulated genes measured by RNA-seq. Values represent the mean ± S.D. (*n* = 3 replicates). (**B**) Drought treatment significantly changes enriched KEGG terms. (**C**) GO terms.

**Figure 7 plants-12-01069-f007:**
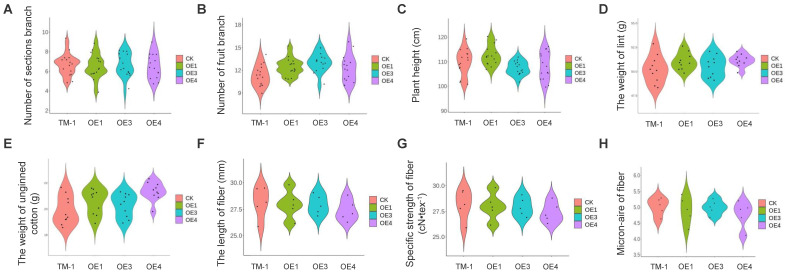
Physiological indexes of TM-1, OE1, OE3, and OE4 plants at various growth periods. (**A**) Number of sections. (**B**) Number of fruits. (**C**) Plant height. (**D**) Weight of lint. (**E**) Weight of unginned cotton. (**F**) Length of fiber. (**G**) Specific strength of fiber. (**H**) Micronaire of fiber.

**Figure 8 plants-12-01069-f008:**
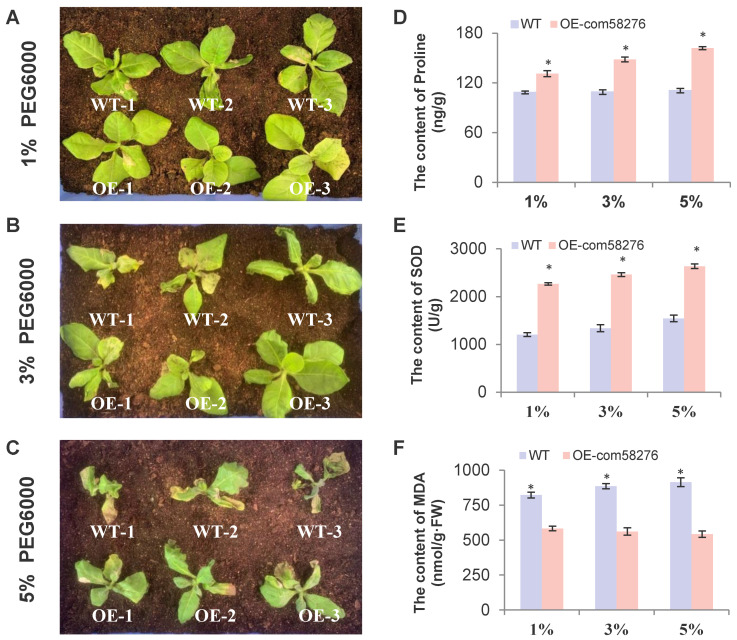
Drought treatment and physiological indicators of overexpressing *com58276* tobacco. (**A**) The phenotype of WT and OE1, OE2, and OE3 tobacco under 1% PEG6000, (**B**) 3% PEG6000, and (**C**) 5% PEG6000. (**D**) Proline activities under treatment. (**E**) SOD activities under treatment. (**F**) CAT activities under treatment. Values are represented by the mean ± S.D. of three biological replicates. * *p* < 0.05 Student’s *t* test.

**Figure 9 plants-12-01069-f009:**
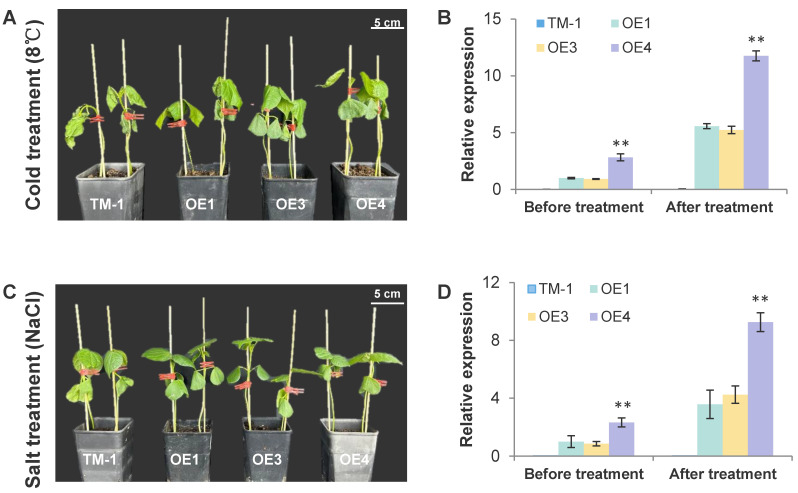
Cold and salt treatment and relative overexpression of *com58276*. (**A**) The phenotype of TM-1 and OE1, OE3, and OE4 cotton under cold treatment (8 °C). (**B**) The expression of *com58276* after cold treatment. Values represent the mean ± S.D (*n* = 3 replicates). ** *p* < 0.01; Dunn’s test. (**C**) The phenotype of TM-1 and OE1, OE3, and OE4 cotton under salt treatment (300 mM NaCl). (**D**) The expression of *com58276* after salt treatment. Values represent the mean ± S.D (*n* = 3 replicates). ** *p* < 0.01; Dunn’s test.

## Data Availability

For privacy reasons, data can be provided via private mail.

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
