# Peer review of "Overexpression of the Caragana korshinskii com58276 Gene Enhances Tolerance to Drought in Cotton (Gossypium hirsutum L.)"

_plants, 2023, doi:10.3390/plants12051069_

Round 1
Reviewer 1 Report
Main issue:
In the manuscript Discussion paragraph is empty.
Other issues:
Generally the Figures are of poor quality - with blurred words, numbers, graphs and pictures.
In some cases you cannot make out what is written
Details:
1. Line 82: The root system of the drought-tolerant lemon desert crop is well developed (Figure.1A)
What is lemon desert crop?
Figure 1A does not show a well-developed root system, I would not describe it that way
2. Line 84. In the 2020 paper (https://doi.org/10.1016/j.gene.2019.144170) when the comp58276 (I believe it should be com58276) was first identified it was described as encoding 84 amino acids, here 85, there is an error here - the stop codon was forgotten
3. Line 86: BLAST analysis on cottonFGD (https://cottonfgd.net/) and NCBI found no homologous genes in cotton, nor similar DNA or protein sequences, demonstrating this is a C.korshinskii-specific gene. We further constructed a genetic transformation vector (Figure.1D) and a subcellular localization vector (Figure.1E) with the DsRed2 tag.
This gene was described in the aforementioned publication and a genetic construct with the DsRed2 tag was obtained then, not in this work. The Authors of the manuscript should refer to the previous paper and not write as if the construct was prepared for this paper. There is no description of the preparation of the construct with the DsRed2 tag in Materials and Methods.
4. Line 90: Subcellular localization results showed that the transcript was localized to the cell membrane and the cytoplasm – Here Authors addressed the question about protein localization not its transcript
5. Figure 1F – should be stated what type of cells are shown on the figure
6. Line 105 – The amplification product was sequenced and aligned to the comp58276 reference genome. – the sentence is wrong – is there comp58276 reference genome?
7. Line 142: SOD and CAT activity, and Pro and MDA content are important physiological 142 indicators of plant stress tolerance. Since mentioned for the first time abbreviation should be explained
8. In the chapter : 2.5. Transcriptome analysis reveals the mechanisms of drought tolerance induced by comp58276 - it would be important to add how many genes were deregulated (up- and down- expressed)
9. Bad quality of the figure 6, lacking description of presented values for RNAseq and qRT-PCR
10. Chapters 3.1 and 3.2 should be included in the results section
11. Line 219 – there should be added the reference to the first publication showing the function of cop58276 gene in Arabidopsis from 2020
12. In the chapter: 3.2. Comp58276 is involved in a variety of adversity stress responses – there is discrepancy with the described numbering on the Figure S2
13. In the Results section, the authors state 300 mM NaCl concentration, whereas in Materials and Methods they state 200 mM NaCl - which is correct?
Editing problems
Line 70: While increasing research efforts were. Employed to identify drought-resistant
Line 191: by comp58276.further 191 scanned root systems and found fewer lateral roots in WT than OE (Figure.4D), as well 192 as significantly lower weight aboveground (Figure.4E) and underground (Figure.4H) in 193 WT.
Author Response
Response to Reviewer 1 Comments
Point 1: In the manuscript Discussion paragraph is empty.
Response 1: The third part of the manuscript is Discussion, Is it because of any version problem that the reviewer did not see it? We included it in the MS submission.
Point 2: Line 82: The root system of the drought-tolerant lemon desert crop is well developed (Figure.1A). What is lemon desert crop?
Response 2: Probably due to a translation error, in order to make it more rigorous, we deleted "lemon"
Point 3: Line 84. In the 2020 paper (https://doi.org/10.1016/j.gene.2019.144170) when the comp58276 (I believe it should be com58276) was first identified it was described as encoding 84 amino acids, here 85, there is an error here - the stop codon was forgotten
Response 3: Thank you very much for reminding you, we have changed all “comp58276” to “com58276”and changed the number of amino acids to “84”.
Point 4: Line 86: BLAST analysis on cottonFGD (https://cottonfgd.net/) and NCBI found no homologous genes in cotton, nor similar DNA or protein sequences, demonstrating this is a C.korshinskii-specific gene. We further constructed a genetic transformation vector (Figure.1D) and a subcellular localization vector (Figure.1E) with the DsRed2 tag.This gene was described in the aforementioned publication and a genetic construct with the DsRed2 tag was obtained then, not in this work. The Authors of the manuscript should refer to the previous paper and not write as if the construct was prepared for this paper. There is no description of the preparation of the construct with the DsRed2 tag in Materials and Methods
Response 4: In order to be able to identify GM crops with DsRed after obtaining positive seedlings, we constructed with reference to the vector of gene paper (https://doi.org/10.1016/j.gene.2019.144170). Thank you very much for the suggestion, we revised it in the manuscript
Point 5: Line 90: Subcellular localization results showed that the transcript was localized to the cell membrane and the cytoplasm – Here Authors addressed the question about protein localization not its transcript
Response 5: Thank you. Revise the “transcript” to “protein”.
Point 6: Figure 1F – should be stated what type of cells are shown on the figure
Response 6: Thank you. We add the detail of “in cotton protoplast” in figure annotations.
Point 7: Line 105 – The amplification product was sequenced and aligned to the comp58276 reference genome. – the sentence is wrong – is there comp58276 reference genome?
Response 7: Very detailed reminder, we refer to the sequence in the gene paper(https://doi.org/10.1016/j.gene.2019.144170).
Point 8: Line 142: SOD and CAT activity, and Pro and MDA content are important physiological 142 indicators of plant stress tolerance. Since mentioned for the first time abbreviation should be explained
Response 8: Add in the MS.
Point 9: In the chapter : 2.5. Transcriptome analysis reveals the mechanisms of drought tolerance induced by comp58276 - it would be important to add how many genes were deregulated (up- and down- expressed)
Response 9: Add in the MS.
Point 10: Bad quality of the figure 6, lacking description of presented values for RNAseq and qRT-PCR
Response 10: We changed the figure 6, now presented values for RNAseq and qRT-PCR will be clearly displayed in the Figure 6
Point 11: Chapters 3.1 and 3.2 should be included in the results section
Response 11: We think that this part of the results deserves more in-depth study and discussion, so we did not put it in the results but in the discussion.
Point 12: Line 219 – there should be added the reference to the first publication showing the function of cop58276 gene in Arabidopsis from 2020
Response 12: Add in the MS.
Point 13: In the chapter: 3.2. Comp58276 is involved in a variety of adversity stress responses – there is discrepancy with the described numbering on the Figure S2
Response 13: Yes, we wanted to show that com58276 is involved in three adversity reactions: drought, low temperature and salt.
Point 14: In the Results section, the authors state 300 mM NaCl concentration, whereas in Materials and Methods they state 200 mM NaCl - which is correct?
Response 14: Yes, It is 300 Mm,changed.
Point 15: Line 70: While increasing research efforts were. Employed to identify drought-resistant
Response 1: Yes, changed.
Point 16: Line 191: by comp58276.further 191 scanned root systems and found fewer lateral roots in WT than OE (Figure.4D), as well 192 as significantly lower weight aboveground (Figure.4E) and underground (Figure.4H) in 193 WT.
Response 16: Yes, changed.

Reviewer 2 Report
Line 38: change “total weight” to “total biomass”
Line 39,40 and 51: word “bell” should be “boll” confirm
Line 70: delete “.Employed” it should be “employed”
Line71: delete “policy or production, whereby the ” rather use “functional traits for”
Line 72 and 73: delete: is an urgent -----and production”
Line 78: change “cultivation” to “development”
Line: 79: add “under dry environment” before full stop to complete sentence
Line 101: Agrobacterium in italics
Line: 107 and 108: what tissues were used for RNA isolation for the northern blot?
Line 219 and 220: Arabidopsis thaliana make it italics
Figure S2 indicate n=3 but how many plants per replicate is not mentioned? Add information if possible.
Line 299 and 300: Agrobacterium tumefaciens make it italics. “ , which was subsequently used for transformation in cotton protoplast”
Line 324: change we Plant – We plan to
Questions:
1) what was the reference gene used for qPCR ?
2) controlled drought experiments with reduced field capacity in the soil are missing?
3) What is probable mechanism and pathway in which this gene is integrated in the cotton genome. Can you give some clues using RNASeq data?
4) Was copy number confirmed in overexpressing lines? Was it made sure that all the OE lines are single copy inserts?
5) What is observed phenotype and possible physiological mechanism (photosynthesis/transpiration/stomatal closure) for the function of OE lines?
Author Response
Response to Reviewer 2 Comments
Point 1:
Line 38: change “total weight” to “total biomass”
Line 39,40 and 51: word “bell” should be “boll” confirm
Line 70: delete “.Employed” it should be “employed”
Line71: delete “policy or production, whereby the ” rather use “functional traits for”
Line 72 and 73: delete: is an urgent -----and production”
Line 78: change “cultivation” to “development”
Line: 79: add “under dry environment” before full stop to complete sentence
Line 101: Agrobacterium in italics
Line: 107 and 108: what tissues were used for RNA isolation for the northern blot?
Line 219 and 220: Arabidopsis thaliana make it italics
Figure S2 indicate n=3 but how many plants per replicate is not mentioned? Add information if possible.
Line 299 and 300: Agrobacterium tumefaciens make it italics. “ , which was subsequently used for transformation in cotton protoplast”
Line 324: change we Plant – We plan to
Response 1: Many thanks to Review1 for the details revise of our MS, we have made all the changes in the manuscript following his suggestions.
Point 2: what was the reference gene used for qPCR ?
Response 2: The cotton gene Histone3 and UBQ7, with stable expression in different tissues, developmental stages, and environmental conditions.(We writed it in the Materials and Methods :RNA extraction, cDNA preparation, and qRT-PCR analyses)
Point 3: controlled drought experiments with reduced field capacity in the soil are missing?
Response 3: Very good suggestion, we have tried drought experiments in the field, but the natural drought in the field is difficult to control, we will try to carry out the drought experiment in the field in Xinjiang, China. The climate there can help us achieve this verification.
Point 4: What is probable mechanism and pathway in which this gene is integrated in the cotton genome. Can you give some clues using RNASeq data?
Response 4: RNASeq data does not seem to give results. I think this should be based on the trans action of Agrobacterium to express protein in the Vir region of the helper plasmid and the T-DNA region of the recombinant plasmid (the target fragment) to activate the transfer of T-DNA, thereby integrating the target fragment into the plant cell genome. With the formation of calluses, the new cells contain the gene of the target fragment. Of course, this doesn't seem to have much to do with our study.
Point 5: Was copy number confirmed in overexpressing lines? Was it made sure that all the OE lines are single copy inserts?
Response 5: Yes,We identified through Southern blot that all three OEs were single-copy insertions.
Point 6: What is observed phenotype and possible physiological mechanism (photosynthesis/transpiration/stomatal closure) for the function of OE lines?
Response 6: This is a good question, we looked at the phenotypes of OE and WT, and there was no difference between the two. And also the photosynthesis and transpiration have no significant differences. Therefore, we think that it is very likely that the function of genes itself confers stress tolerance on plants.

Round 2
Reviewer 1 Report
Unfortunately most of the figures are still blurred. It is not possible to see KEGG terms and (C) GO terms on the Figure 6.
Author Response
2nd Response to Reviewer 1 Comments
Point 1: Unfortunately most of the figures are still blurred. It is not possible to see KEGG terms and (C) GO terms on the Figure 6.
Response 1: Thank you very much for your suggestion, we have revised the clarity of the figure again, and, as a supplement, we will provide our original figure directly to the editorial office so that they can choose it during the final layout process.

Round 3
Reviewer 1 Report
Dear Authors,
I have received the file with the figures, everything looks very good.
Congratulations